# Holistic-CAM: Ultra-lucid and Sanity Preserving Visual Interpretation in Holistic Stage of CNNs

## ABSTRACT

As the visual interpretations for convolutional neural networks (CNNs), backpropagation attribution methods have been garnering growing attention. Nevertheless, majority of those methods merely concentrate on the ultimate convolutional layer, leading to tiny and concentrated interpretations that fail to adequately clarify the model-central attention. Therefore, we propose a precise attribution method (i. e., Holistic-CAM) for high-definition visual interpretation in the holistic stage of CNNs. Specifically, we first present weighted positive gradients to guarantee the sanity of interpretations in shallow layers and leverage multi-scale fusion to improve the resolution across the holistic stage. Then, we further propose fundamental scale denoising to eliminate the faithless attribution originated from fusing larger-scale components. The proposed method is capable of simultaneously rendering fine-grained and faithful attribution for CNNs from shallow to deep layers. Extensive experimental results demonstrate that Holistic-CAM outperforms state-of-the-art methods on common-used benchmarks, including deletion and insertion, energy-based point game as well as remove and debias on ImageNet-1k, it also passes the sanity check easily.

## CCS CONCEPTS

• **Computing methodologies** → **Computer vision**.

## KEYWORDS

Visual Interpretation, Class Activation Map, Multi-scale Fusion

## 1 INTRODUCTION

Deep Neural Networks (DNNs) have achieved unprecedented break through in a variety of single-modal and multi-modal tasks, such as image classification [1], object detection [2], visual reasoning [3], diffusion model [4] and video recognition[5]. To establish trust in these models, it is crucial to comprehend and articulate the operational procedures and reasoning behind the decision-making process of the models in a lucid and understandable manner [6].

Currently, various methods are proposed to interpret media-related models, especially Class Activation Map (CAM) [7]. As the visual interpretations for convolutional neural networks (CNNs), CAM and its derivative methods [8–13] employ weighted linear

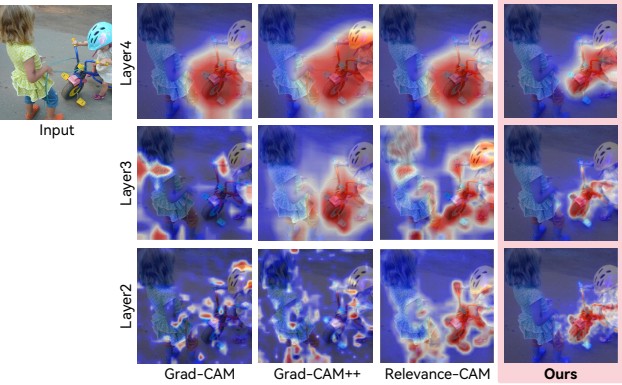

**Fig 1.** Visualization result of Grad-CAM, Grad-CAM++, Relevance-CAM and proposed Holistic-CAM. The label of the input is tricycle. Holistic-CAM generates faithful and fine-gained from deep to shallow layers.

summations of the ultimate layer activation maps to generate faithful and robust interpretations (or attributions). Nevertheless, there is a significant drawback that those methods tend to generate extremely narrow interpretations at very low resolution in deeper layers [14–17]. Meanwhile, these methods also incur messy and faithless attributions in shallow layers, as shown in "Layer2" and "Layer3" in Fig. 1. This drawback can be attributed to two primary factors. One is associated with the gradient loss derived from the non-linear activation functions such as Rectified Linear Unit (ReLU) and Sigmoid, as well as the frequently-used channel-wise weights that fail to assign correct importance onto the forward features in the shallow layers [14, 15]. The other is linked to the low-resolution feature space of deep layers, which directly restricts the traditional CAM methods to provide high-definition interpretations [16].

Previous researchers have endeavored to address these limitations. One group of studies utilize the unaffected positive gradient [14, 15] or layer-wise relevance [18] to provide robust interpretations in shallow layers. Nonetheless, their practical impacts remain inadequate in output layers, which usually provide vague interpretations (e.g. Relevance-CAM in Fig. 1) and fail to plainly elucidate the model's focal attention. Others systematically perform multi-scale accumulation and fusion of the activation maps as well as backpropagated gradients to enhance resolution in output layers [16, 17]. However, they frequently encounter unfaithful attributions stemming from the fusion of larger-scale components. This will compromise the credibility and robustness of the interpretations and potentially lead to misunderstandings, which steers the interpreting process in a contradictory direction. Thus, it is imperative to seek potential solutions to attain the pinnacle of enhanced resolution and interpretive precision, ultimately offering a ultra-lucid and faithful approach to aid humans to comprehend those models.

To address the challenges above, we propose Holistic-CAM, a novel attribution method to provide high-resolution and sanity-preserving visual interpretation throughout the holistic stage of

CNNs. Specifically, we firstly present a weighted positive gradient-based weight to maintain the clarity and robustness of interpretations in shallow layers. Then we leverage a multi-scale fusion scheme to guarantee the high-resolution within the holistic stage. Ultimately, we extract the fundamental scale map and utilize it to examine the faithfulness attributions and enhance the comprehensive fidelity of interpretations. As presented in Fig. 1, Holistic-CAM outperforms other methods of visualizing target objects.

The contributions of this paper are as follows:

- We propose a precise attribution method (i. e., Holistic-CAM) based on the proposed weighted positive gradients and multi-scale fused features. It is capable of providing high-definition interpretations within the holistic stage of CNNs.
- We present a fundamental scale denoising strategy based on the proposed low-pass wrap. It eliminates the faithless attributions and preserves the holistic fidelity of Holistic-CAM.
- We conduct extensive visual assessments and quantitative evaluations on Holistic-CAM with other state-of-the-art methods, experimental results illustrate that Holistic-CAM exhibits remarkable performance on the evaluations of holistic fidelity and localization abilities.

## 2 PRELIMINARIES AND MOTIVATION

**Problem Statement.** Consider a pre-trained convolutional neural networks as a function $f : \mathcal{X} \to \mathbb{R}^K$ of an input image $x \in \mathcal{X} \subseteq \mathbb{R}^{w \times h}$ with softmax output of $K$ classes as $f_c(x) \geq 0$ for $c = 1, ....., K$ and $\sum_{c=1}^{K} f_c(x) = 1$. We aim to attribute the relevance between input image $x$ and target class $c$, ultimately producing an attribution map $M(x) \in \mathbb{R}^{w \times h}$.

**Notations.** In CAM-based attribution methods, we regard $f_c$ as a composite function, i. e., $f_c(x) = g_c \circ A(x)$, where $A(x)$ is the activation map of a certain layer w.r.t. the specific input $x \in \mathcal{X}$, and $g_c$ is the layers between $A(x)$ and the output. In addition, we denote $A_{ij}^k$ as the $(i, j)$-th neuron activation values in the $k$-th activation map $A(x)$, and denote $G_{ij}^{kc}$ as the gradient between prediction score $f_c$ and input feature $A_{ij}^k$.

**Background: Gradient-based CAMs.** As the CAM-based attribution method, Grad-CAM [8] is undoubtedly most recognized. It generate interpretation with the channel-wise importance weight $w_k^c$ based on the spatial average of the gradient of $f_c(x)$, i. e.,

$$w_{Grad-CAM}^{kc} = GAP\left(\frac{\partial f_c(x)}{\partial A^k}\right) \qquad (1)$$

$$M_{Grad-CAM}^c(x) = ReLU\left(\sum_k w_{Grad-CAM}^{kc} \cdot A^k(x)\right) \qquad (2)$$

Motivated by its limited capability to identify multiple occurrences and pinpoint entire objects within adversarial samples precisely, Grad-CAM++ [9] introduced an element-wise weight $\alpha_{ij}^{kc}$ to enhance the localization ability of positive gradients. To be specific, they revised the approach of seeking channel-wise importance as:

$$w_{Grad-CAM++}^{kc} = GAP\left(\alpha^{kc} \cdot ReLU(\frac{\partial f_c(x)}{\partial A^k})\right) \qquad (3)$$

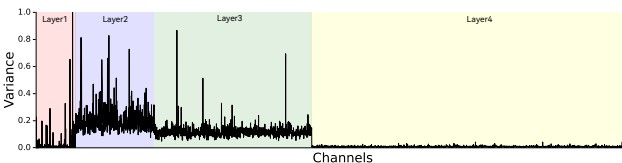

**Fig 2. Numeral Analysis on the Gradient.** In the deep stage (i.e. Layer4), the average gradient variances of individual channels approach zero. Conversely, in the shallow stages (Layer3 to Layer1), the gradient variances are more pronounced.

For a CNN with a GAP layer, the final classification score $f_c$ for a certain predicted result $c$ can be written as a linear combination of the output in average pooled features [8]

$$f_c(x) = \sum_k w_{Grad-CAM++}^{kc} \cdot \sum_i \sum_j A_{ij}^k \qquad (4)$$

To obtain the positive gradient weight $\alpha_{ij}^{kc}$, two partial derivatives between classification score $f_c$ and individual neuron activation $A_{ij}^k$ are performed. The gradient weight $\alpha_{ij}^{kc}$ is obtained after rearranging the form:

$$\frac{\partial^2 f_c(x)}{(\partial A_{ij}^k)^2} = 2 \cdot \alpha_{ij}^{kc} \cdot \frac{\partial^2 f_c(x)}{(\partial A_{ij}^k)^2} + \alpha_{ij}^{kc} \cdot \frac{\partial^3 f_c(x)}{(\partial A_{ij}^k)^3} \cdot \sum_a \sum_b A_{ab}^k \qquad (5)$$

$$\alpha_{ij}^{kc} = \frac{\frac{\partial^2 f_c(x)}{(\partial A_{ij}^k)^2}}{2\frac{\partial^2 f_c(x)}{(\partial A_{ij}^k)^2} + \frac{\partial^3 f_c(x)}{(\partial A_{ij}^k)^3} \sum_a \sum_b A_{ab}^k} \qquad (6)$$

here, $(i, j)$ and $(a, b)$ are iterators over the same activation map $A^k$, which is set to prevent confusion. Finally, the attribution map of this method is generated by integrating modified channel-wise weight w.r.t. activation maps:

$$M_{Grad-CAM++}^c(x) = ReLU\left(\sum_k w_{Grad-CAM++}^{kc} \cdot A^k(x)\right) \qquad (7)$$

**Observation I: Incorrect interpretations of utilizing channel-wise weights.** The utilization of channel-wise importance weights $w^{kc}$ in Eq. 1 and Eq. 3 is inspired by the GAP layer in the CAM method [7], which has since been adopted by other techniques such as Score-CAM [12] and Relevance CAM [18]. During introducing gradient-based channel-wise weights [8, 14, 16] or other forms [12, 18], class activation mapping has expanded interpretative capabilities beyond the constraints of the last convolutional layer, and reached any layer. Nonetheless, shallower layers have larger and less dispersed feature spaces where the channel-wise weights struggle to accurately capture differentiated positional information [14]. In this case, the element-wise weighting strategy seems more reasonable.

Here, we conducted a simple numerical analysis on the gradients of ResNet-50 across its four stages (Layer 1 to Layer 4) to thoroughly examine the disparities between channel-wise weights and element-wise weights. As depicted in Fig. 2, the variances of most activation maps approach zero in the final stage. In this case, the distinction between global weights and individual element weights are nearly identical. However, in earlier stages, the variances of most feature maps are notably high, posing challenges for global gradient-based weights to provide effective interpretations in shallower layers. In

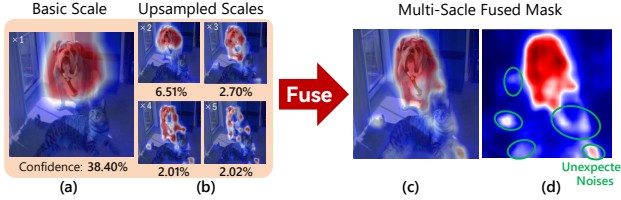

**Fig 3. Noise in the Multi-scale Fusing Progress.** The enlarged scales are marked in the upper left corner of (a) and (b) and the corresponding classification confidence is indicated below.

summary, utilizing element-wise weight is a more efficient approach for elucidating the holistic stage of CNNs than channel-wise weight.

**Question I: Are the channel-wise importance weights always rational?** With this in mind, the impact of the element-wise weighting coefficient $\alpha_{ij}^{kc}$ (i. e., Eq. 3), which amplifies the utilization of solely positive gradients to replicate the results obtained through complete gradient utilization, may be ineffective due to the superstition on channel-wise weighting (i. e., Eq. 7). So, is there any solution to fully realize its genuine value?

**Observation II: Unreliable attributions of multi-scale fusion methods.** In the domain of computer vision, multi-scale fusion and image pyramids are established techniques for augmenting informational capacity [19]. These approaches leverage the flexibility of processing inputs at various scales, thereby enabling the extraction of a more comprehensive set of features from a larger data space [20]. With regard to attribution methods, although multi-layer fusion [21] or multi-scale input integration technology [16, 17] have been successfully employed to refine the resolution of interpretation, they are constrained by the low fidelity of high-resolution input, potentially leading to unreliable attribution maps as the confidence in the data is sub-optimal.

**Question II: Is it feasible to guarantee both high resolution and fidelity?** A simple example is depicted in Fig. 3. We upscale an image by 2-5 times the original dimensions and document the corresponding attribution maps. Subsequently, integrating those attribution maps across various scales. The result is depicted in Fig. 3 (b). Higher resolutions yield greater detail but also introduce significant noise, which always exists and even be amplified during the fusion process. In addition, although the clarity of the fundamental-scale attribution maps are insufficient, they are generated under the context of high confidence which guarantees enough fidelity. So, can basic-scale masks be employed to optimize larger-scale masks?

## 3 PROPOSED METHOD

In this section, we present a detailed illustration of Holistic-CAM. As depicted in Fig. 5, we utilize a multi-scale fusion scheme to enhance the resolution of generated attributions. Our method comprises three key components: 1) positive gradient enhancement; 2) high-resolution attribution generation; 3) fundamental scale denoising based on low-pass wrap.

### 3.1 Positive Gradient Enhancement

Modern neural networks tend to set up multiple ReLU functions to increase the non-linear fitting capability [22]. However, it inevitably leads to gradient vanishing problems, such as zero-gradient.

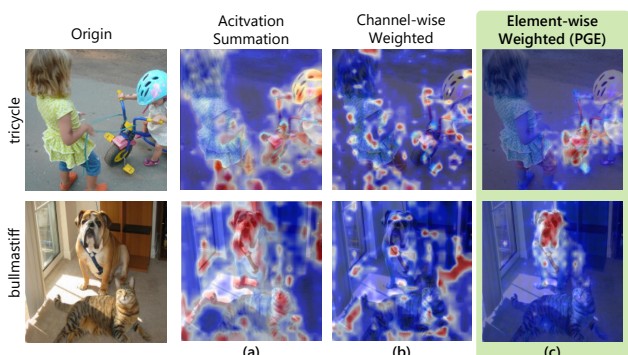

**Fig 4. Gradient Weighting Comparison.** Based on the element-wise weighting strategy, PGE is capable of interpreting correct significance in shallow layers.

Some predecessors [14, 15] prove that using positive gradients helps to obtain robust and fidelity interpretations in shallow layers. Drawing upon this background and the discussed **Observation 1**, we put forward the method to fully realize the inherent and genuine value of positive gradient within shallow layers, namely *Positive Gradient Enhancement (PGE)*.

In regard to the gradient $G_{ij}^{kc}$ from a specific layer $l$, the PGE operates by allocating the positive gradient weight $\alpha$ from equation 6 through element-wise weighting:

$$PGE(G_{ij}^{kc}) = \alpha_{ij}^{kc} \cdot ReLU(G_{ij}^{kc}) \tag{8}$$

$$= \frac{\frac{\partial^2 f_c(x)}{(\partial A_{ij}^k)^2}}{2\frac{\partial^2 f_c(x)}{\partial (A_{ij}^k)^2} + \frac{\partial^3 f_c(x)}{(\partial A_{ij}^k)^3}\sum_a\sum_b A_{ab}^k + eps} \cdot relu(\frac{\partial f_c(x)}{\partial A_{ij}^k})$$

To be noticed, weighting is meaningless where the one-order gradient is zero. To address this issue, here, a bias term of $eps = 10^{-5}$ is incorporated to prevent the occurrence of zero denominators.

To gain a more intuitive comprehension, we sample attribution maps obtained by different weighting strategies in the intermediate stage (Layer2) of ResNet-50, i.e. *(a) Activation Summation:* $W_{ij}^{kc} = 1$, *(b) Channel-wise Weighting:* $W_{ij}^{kc} = w^{kc} = \sum_k \alpha_{ij}^{kc} \cdot ReLU(G_{ij}^{kc})$, *(c) Element-wise Weighting:* $W_{ij}^{kc} = PGE(G_{ij}^{kc})$. In a conceptual sense, (b) and (c) correspond to the results after applying channel-specific weights or individual element weights onto the activation summation, denoted as (a). Moreover, the attribution maps of those methods are calculated in uniform specifications, as $M_{ij}^c = \sum_k W_{ij}^{kc} \cdot A_{ij}^{kc}$.

Detailed results are depicted in Fig. 4. Due to assigning correct importance to neurons, PGE contributes to generating robust and faithful interpretations within shallow layers. On the contrary, the utilization of channel-wise weights not only fails to generate faithful interpretations but also disrupts the integrity of the original features in Fig. 4 (a).

### 3.2 High Resolution Attribution Generation

In this subsection, we introduce the process of generating our holistic-stage high-resolution attribution based on the aforementioned *Positive Gradient Enhancement*. Specifically, we firstly iteratively interpolate the original input image $I^{\zeta_0}$ into $T$ scales

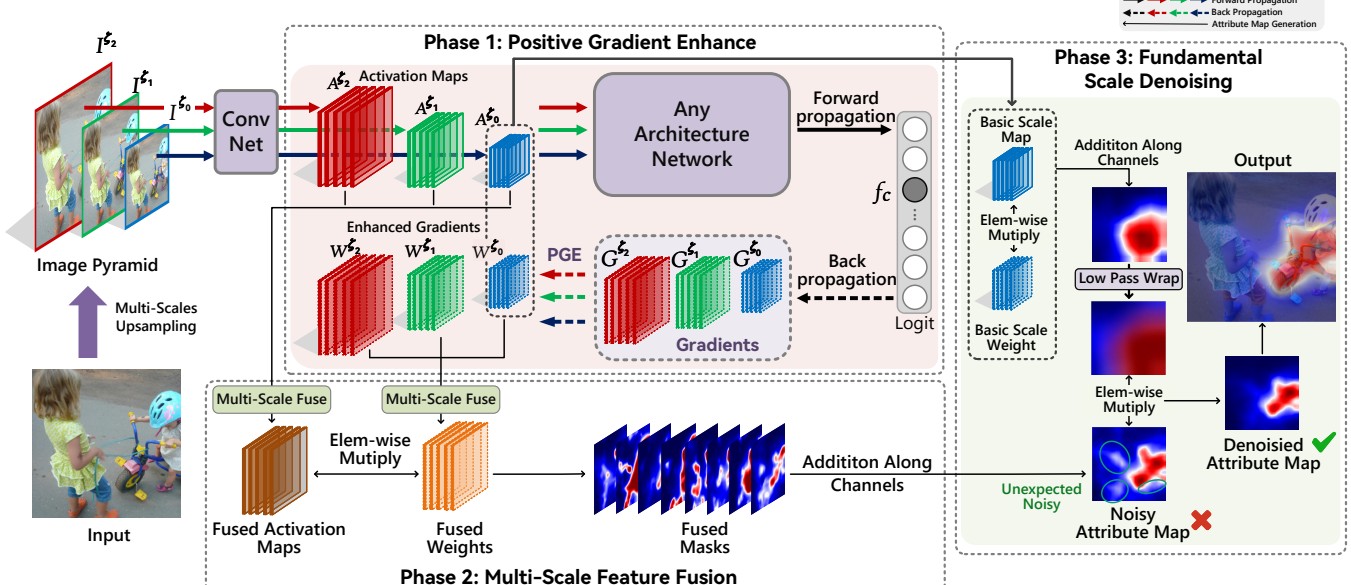

**Fig 5. Pipeline of Holistic-CAM.** The initial input is transformed into an image pyramid, which is utilized to obtain activation maps and enhanced gradients at different scales throughout the forward and backward propagation stages. Combining them produces the primary attribution map, which includes both positive information and unexpected noises originating from various scales. The ultimate attribution map is derived following the implementation of the denoising process on the fused attribution map.

$\zeta_1, \zeta_2, ..., \zeta_T$, i. e., $I^{\zeta_1}, I^{\zeta_2}, ..., I^{\zeta_T}$. Here, the interpolation function is represented as $\varphi(I^{\zeta_0}, \zeta_t)$, which utilizes bi-linear interpolation to resize the resolution of $I^{\zeta_0}$ from $\zeta_0$ to $\zeta_t$:

$$I_{\zeta_t} = \varphi(I^{\zeta_0}, \zeta_t),$$
$$\zeta_t = \zeta_0 + \frac{\zeta_T}{T}(t-1) \tag{9}$$

here $\zeta_T$ represents the maximum resolution threshold, $T$ indicates the total iterations.

Then, we transform the multi-scale images in the form of image pyramid to be the inputs of thenetwork, then record the forward activation $A^{\zeta_0}, A^{\zeta_1}, ..., A^{\zeta_T}$ and the corresponding gradient $G^{\zeta_0}, G^{\zeta_1}, ..., G^{\zeta_T}$ from a certain layer $l$. After that, we integrate activation maps of different scales:

$$\bar{A} = \frac{1}{T}\sum_{t=1}^{T} A^{\zeta_t} \tag{10}$$

In each iteration, we compute the enhanced gradients $W^{\zeta_t}$ based on the backward gradients $G^{\zeta_t}$ and the proposed $PGE()$. Then the fusion of these enhanced gradients can be obtained by:

$$W^{\zeta_t} = PGE(G^{\zeta_t})\bar{W} = \frac{1}{T}\sum_{t=1}^{T} W^{\zeta_t} \tag{11}$$

Formally, the primary high-resolution attribution map $M_c^{PHR}$ is generated after assigning the element-wise weight $\bar{W}$ onto the fused activation maps $\bar{A}$ and accumulating along the dimension of channel:

$$M_c^{PHR} = \sum_k \bar{W}^k \odot \bar{A}^k \tag{12}$$

where $k$ represents the iterator of channels and $\odot$ denotes element-wise multiply operation.

## 3.3 Fundamental Scale Denoising

In response to the aforementioned **Observation 2** which aims to identify a solution that retains the original high-resolution details and eliminates low-fidelity information. Therefore, a *Fundamental Scale Denoising (FSD)* method is put forward based on later proposed *low-pass wrap*.

As illustrated in the *Phase* 3 of Fig. 5, the low-resolution salient regions demonstrate stronger localization capabilities than the maps with higher resolutions, but they tend to lose fine details due to the limited resolution; Furthermore, the basic-scale attribution maps typically contain a significant amount of high-frequency details. Integrating them directly with the fused attribution maps will undoubtedly compromise the distinctive details inherent to higher resolution. To address these issues, we propose a low-pass wrap (LPW) strategy aimed at eliminating noise beyond the localization area without compromising the high-definition information of high resolution.

Specifically, given a multi-scale fused map $M_c^{fused}$, we first separate its fundamental scale component $M^{\zeta_0}$. To wrap the low-frequency component, we subsequently filter out the high-frequency information that exceeds the mean value $\bar{M}$:

$$M_{ij}^{filted} = \begin{cases} \bar{M}, & if \quad M_{ij}^{\zeta_0} > \bar{M}; \\ M_{ij}^{\zeta_0}, & otherwise. \end{cases} \tag{13}$$

where $\bar{M} = (max(M^{\zeta_0}) + min(M^{\zeta_0}))/2$.

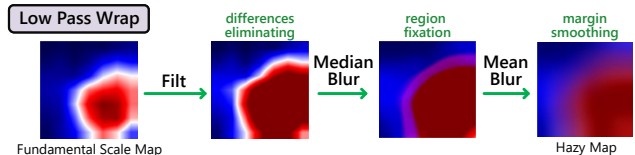

**Fig 6. Overview of Low-pass Wrap (LPW)**. LPW is used to maintain the precision of regional positioning, it contributes FSD to eliminate noise through the multiplication of the multi-scale fused map and its output.

Although fixed value filtering is capable of determining the positioning area without discrimination, it fails to preserve the marginal details of larger-scale maps. Otherwise, the fundamental scale map contains fewer details and less semantic information compared to the multi-scale fused map. To address this issue, we employ a median filter ($median\_blur2d$) to mitigate high-frequency information of the edge and reasonably expand the positioning area:

$$M^{median} = median\_blur2d(M^{filted}, blurSize) \qquad (14)$$

Finally, a mean filter ($mean\_blur2d$) is applied to strengthen the smoothness of the margin zone, that is:

$$M^{LPW} = mean\_blur2d(M^{median}, (ksize, ksize)) \qquad (15)$$

To avoid confusion, here the superscripts $filted$ and $median$ denote the different processing phases of fixed value filtering and median filtering. Finally, the hazy base-scale mask $M^{LPW}$ owns plentiful low-pass content, which indicates the intact localization information of the target object. The integral process of fundamental scale denoising can be summarized as:

$$FSD(M_c^{fused}) = M_c^{fused} \odot M^{LPW} \qquad (16)$$

where $\odot$ denotes element-wise multiply operation.

### 3.4 Holistic-CAM

Our Holistic-CAM involves the joint effort of all the aforementioned components. With regard to the primary high-resolution map $M_c^{PHR}$ obtained in subsection 3.2, FSD is necessary to prevent the occurrence of unfaithful attributions and guarantee the sanity of interpretations.

Specifically, Holistic-CAM is obtained by executing *Fundamental Scale Denoising* onto the fused attribution map:

$$M_c^{Holistic-CAM} = FSD(M_c^{PHR}) \qquad (17)$$

## 4 EXPERIMENTS

In section 5, we evaluate our causal attribution method with state-of-the-art methods via experiments in diverse forms, including qualitative visualization evaluation, quantitative evaluation, ablation study and saliency check.

### 4.1 Experimental Setups

**Datasets and Backbone**. Experiments in this section are conducted on the validation split of ImageNet-1k [23] (containing 50k images). All the images are resized to $3 \times 224 \times 224$, and transformed to tensors after normalized. We utilize pre-trained torch-vision models VGG-16 and ResNet-50 as the backbone. For a fair comparison, attribution maps are up-sampled through bi-linear interpolation to $224 \times 224$. All the experiments are carried on one NVIDIA RTX 3090 GPU.

**Baselines**. We set up comparisons with state-of-the-art CAM-based attribution methods, such as Grad-CAM [8], Grad-CAM++[9], XGrad-CAM[10], Eigen-CAM[11], Layer-CAM[14], CAMERAS[16] and Relevance-CAM[18].

**Evaluation Metrics.** We utilize *Deletion and Insertion* as well as *Remove and Debias* metrics to evaluate the holistic fidelity of different attribution methods. Meanwhile, we also conduct *Energy-based Pointing Game* to evaluate the localization abilities.

**1) Deletion and Insertion:** proposed by [24]. It contained three pixel-level fidelity metrics, i.e. *Deletion, Insertion* and *Over-all*. *Deletion (Del)* measures the rate of classification confidence decreases as the elements are deleted in order of importance according to the attribution map, *lower is better*. *Insertion (Ins)* measures the rate of classification confidence increases as the elements are inserted in order of importance, *higher is better*. All the results are expressed through the Area Under the Curve (AUC) where the horizontal axis represents the percentage of elements deleted or inserted, and the vertical axis represents the pre-softmax classification probability. *Over-all score*, integrates the deletion and insertion results as *AUC(Insertion)-AUC(Deletion)*. Comprehensively assessing the holistic fidelity of each interpretation, *higher is better*.

**2) Remove and Debias (ROAD)**: proposed by [25], a faithful feature attribution metric. At first, *ROAD* utilizes two sorting strategies, i.e., Most Relevant First (MoRF) and Least Relevant First (LeRF) to arrange pixels based on the significance determined by the attribution maps. Subsequently, determine a certain removal ratio $[t_1, t_2, ..., t_N]$ to calculate the fluctuation of classification confidence after removing the most or least relevant features, i.e., $LeRF_{t_i} = (f_c^{LeRF,t_i}(x) - f_c(x))/f_c(x)$ and $MoRF_{t_i} = (f_c(x) - f_c^{MoRF,t_i}(x))/f^c(x)$. Finally, the ROAD score is calculated by integrating LeRF and MoRF scores across different removal ratios, as $ROAD = \sum_{i=1}^{N}(LeRF_{t_i} - MoRF_{t_i})/N$, *higher is better*.

**3) Energy-based Pointing Game (EPG):** Localization ability holds significant importance, particularly in the context of utilizing attribution maps for localization tasks [14, 18]. EPG is initially introduced by [26] and later refined by [12]. It focuses on quantifying the amount of energy derived from the attribution map that is directed towards the designated target area. In particular, the input image undergoes a binarization process based on the established bounding box of the target entity: the interior section is designated a value of 1, while the exterior portion is assigned a value of 0. Then multiply this binary matrix with the generated attribution map, and sum over to count how much energy is in the target bounding box. Generally, this metric can be denoted as:

$$Proportion = \frac{\sum M_{(i,j) \in bbox}^c}{\sum M_{(i,j) \in bbox}^c + \sum M_{(i,j) \notin bbox}^c} \qquad (18)$$

**Implement Details**. **1) Metric Setups:** In *fidelity evaluation*, we randomly sample 3,000 images from ImageNet-1k validation dataset. For *deletion and insertion* metric, 3.6% ($224 \times 8$) pixels of the original image are removed or inserted in each iteration. In regard to *ROAD*, the removal ratio is set as $t_i \in [20\%, 40\%, 60\%, 80\%]$.

To gauge the localization capabilities of the proposed attribution method more accurately, we randomly sampled 5,000 images in the *localization evaluation*. Meanwhile, we consider the coordinates

of the top 100 points within the attribution map instead of solely focusing on the highest value point as [26] for *energy-based point game.*

**2) Parameter Setups:** The hyper-parameter for *low-pass wrap* is determined as *blur_size=51, ksize=91* based on the findings of the following ablation study. Furthermore, we set the maximum resolution of the multi-scale fusion process as $\zeta_T = (1K, 1K)$ after 7 iterations of upsampling, referring to [16, 17].

## 4.2 Qualitative Evaluation

The results of our method are qualitatively assessed against state-of-the-art CAM-based attribution methods (Grad-CAM, Grad-CAM++, XGradCAM, EigenCAM, LayerCAM, CAMERAS, Relevance-CAM) applied to a ResNet-50 model, as shown in Fig. 7. Attributions of randomly chosen images are displayed to showcase the superior precision and faithfulness of Holistic-CAM maps.

Noteworthy is the observation that, in contrast to other methods, Holistic-CAM not only effectively eliminates extraneous noise from other target objects (e.g., "Kuvasz" and "Bull mastiff") but also demonstrates the capability to maintain high resolution within shallow and deep layers simultaneously. In the case of multiple occurrences of the same class (e.g., "Warplane"), the attribution maps generated by Holistic-CAM effectively highlight the significant features of each object, while others make different objects forfeit their independence. Generally, these excellent performances can be attributed to two aspects. One is related to the effort of enhanced positive gradients, which can mitigate the issue of gradient incomplete in shallow layers thereby enhancing the robustness in shallow layers. The other involves the integration of the multi-scale fusion technique as well as the proposed denoising module, which effectively enhances the resolutions of interpretations and simultaneously prevents the occurrence of unfaithful attributions.

In contrast, alternative approaches aimed at improving interpretation accuracy have faced notable difficulties with fidelity. Especially in Relevance-CAM, it functions as an edge detector for recognizing "Bull mastiff" and "Warplane". Even completely ineffective in identifying "Kuvasz". Consequently, Holistic-CAM is proficient in delivering precise and reliable visual interpretations for the complete operation of contemporary deep visual classifiers (e.g., "Tench").

## 4.3 Quantitative Evaluation

Quantitative assessments are carried out to evaluate the faithfulness and localization ability of the interpretations produced by Holistic-CAM across various layers of CNNs. The results of *Ins, Del,* and *ROAD*, presented in Tab. 1, demonstrate that the proposed Holistic-CAM method outperforms other methods on holistic fidelity across the entire stage of VGG-16 and ResNet-50 models. This is mainly attributed to the effort of fusing the multi-scale features that ensure the clarity of interpretations across the holistic stages. Meanwhile, the application of PGE makes an effort to guarantee robust performance in the shallow layers.

Furthermore, our method exhibits superior localization ability, as evidenced by the *EPG* score in Tab. 1. This is primarily attributed to the introduction of FSD, which not only integrates the benefits of the fusion of different multi-scale features to preserve the high-resolution of interpretations but also suppresses the occurrence of unfaithful attribution beyond the intended positioning zone.

## 4.4 Ablation Study

We conduct ablation studies on the primary module *fundamental scale denoising* (F) and *positive gradient enhancement* (P) through visual evaluation and quantitative analysis. In addition, we also conduct parameter study and module analysis on *fundamental scale denoising*, which is presented in the appendix.

**Visualization Evaluation.** Visualization result is presented in Fig. 8 . On the one hand, PGE significantly contributes to enhancing the thoroughness and resilience of the interpretation processes at the shallow layers. On the other hand, FSD efficiently eliminates extraneous noise beyond the designated target region across all stages. It is capable of enhancing the accuracy and reliability of the interpretations.

**Quantitative Evaluation.** We report the quantitative ablation study results on the shallow layer (Layer2) as well as the deep layer (Layer4) of ResNet-50. Concretely, we randomly selected 2,000 images from ImageNet-1k-val-dataset and employed *Del and Ins* and *EPG* metrics to evaluate the contributions of each module on holistic fidelity and localization ability. Detailed result can be indicated in Tab. 2.

In deep layers, the implementation of FSD leads to a notable enhancement in the *Ins, Over-all* and *EPG* scores, indicating a significant improvement on robustness and precision of attribution maps by effectively eliminating the faithless attribution. In addition, we found nearly consistent performance after removing/retaining positive gradient enhancement. This mainly attributed to the absence of significant gradient loss in the deep layer where the effect of positive gradient enhancement is slightly smaller. In shallow layers, although FSD has the potential to enhance feature loyalty, its ability to improve localization accuracy remains limited. On the other hand, there has been a notable enhancement in localization precision following the implementation of PGE. However, its effort on feature loyalty remains restricted. In addition, it is observed that both the feature expression capability and localization accuracy have reached peak levels after integrating FSD and PGE. Therefore, these components in our proposed pipeline ensure each other and that the generated attribution map is not only faithful enough but also maintains robust localization ability.

## 4.5 Saliency Check

As indicated by [27], interpretation methods have the risk of functioning as edge detectors when they solely rely on visual assessment. Therefore, we conduct a saliency check and evaluate our methods with cascading randomization and independent randomization [27]. Fig. 9 (a) is Holistic-CAM result for the VGG-16 model obtained by progressively randomizing the model parameters from logit to Conv19. And Fig. 9 (b) indicates the result obtained by individually randomizing the parameters of each layer. It's obvious that the attribution maps are destroyed along the parameter randomization procedure. Thus, Holistic-CAM is sensitive to model parameters.

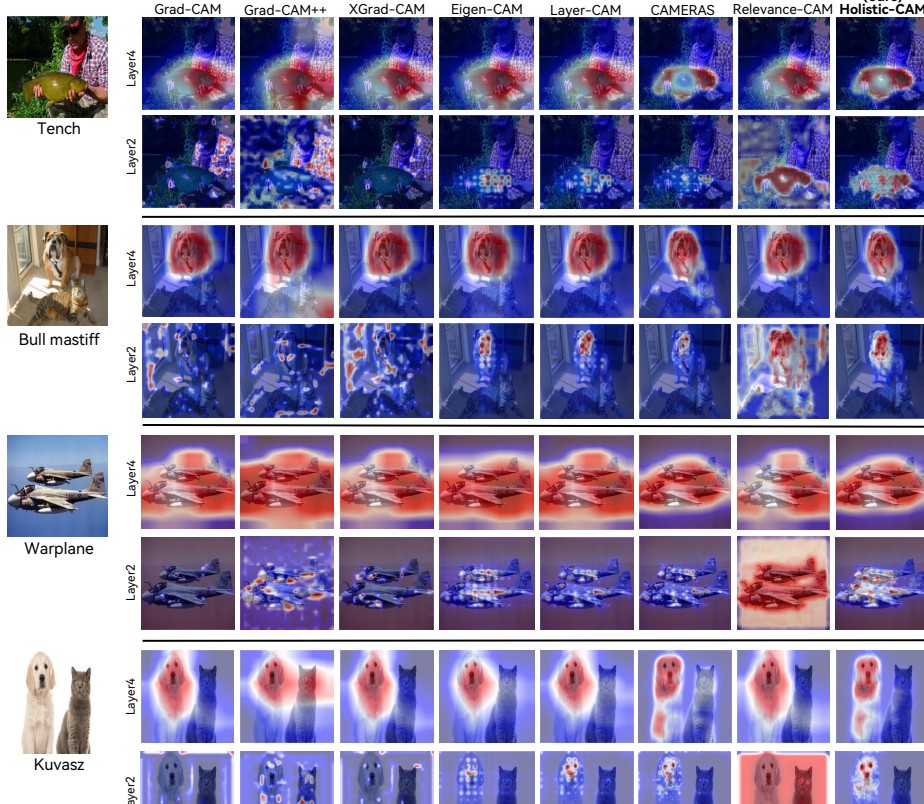

**Fig 7. Comparisons of Various Attribution Methods.** The columns are divided by the interpretation methods. The rows are divided along layer depth. Layer2 represents the intermediate layer and Layer4 represents the last convolutional layer. Holistic-CAM is capable of locating the target object accurately.

**Table 1: Quantitative Comparisons Result** on ResNet-50 and VGG-16. Layer4 and Layer2 represent the last convolutional layer and the intermediate layer in ResNet, respectively. Layer43 denotes the 5th and last max-pooling layer of VGG-16, and Layer23 denotes the 3th max-pooling layer. The best results for each metric are shown in underline bold as well as dyed in **red**, and the second one is shown with underline and colored in blue.

| | Method | Ins ↑ | Del ↓ | Over-all ↑ | ROAD ↑ | EPG ↑ | | Method | Ins ↑ | Del ↓ | Over-all ↑ | Road ↑ | EPG ↑ |
|---|---|---|---|---|---|---|---|---|---|---|---|---|---|
| | | | | ResNet50 | | | | | | | VGG-16 | | |
| Layer4 | Grad-CAM | 54.887 | 11.622 | 43.265 | 28.095 | 53.673 | Layer43 | Grad-CAM | 48.390 | 10.894 | 37.496 | 25.851 | 49.730 |
| | Grad-CAM++ | 51.165 | 14.172 | 36.993 | 22.470 | 51.424 | | Grad-CAM++ | 45.044 | 12.528 | 32.516 | 22.331 | 52.428 |
| | XGrad-CAM | 54.887 | 11.622 | 43.265 | 28.099 | 53.673 | | xGrad-CAM | 49.029 | 10.761 | 38.268 | 26.223 | 49.288 |
| | Eigen-CAM | 53.249 | 12.705 | 40.544 | 25.595 | 53.167 | | Eigen-CAM | 48.925 | 10.547 | 38.378 | 25.765 | 52.305 |
| | Layer-CAM | 54.018 | 11.882 | 42.136 | 26.799 | 52.963 | | Layer-CAM | 48.125 | 10.444 | 37.681 | 26.002 | 51.433 |
| | CAMERAS | 54.439 | 8.698 | 45.741 | 28.606 | 52.931 | | CAMERAS | 44.548 | 9.091 | 35.457 | 26.153 | 50.008 |
| | Relevance-CAM | 54.663 | 11.622 | 43.041 | 27.981 | 52.989 | | Relevance-CAM | 49.296 | 10.043 | 39.253 | 25.98 | 50.894 |
| | **Holistic-CAM** | **55.056** | **8.947** | **46.109** | **29.047** | **57.635** | | **Holistic-CAM** | **49.569** | **8.792** | **40.777** | **26.345** | **55.090** |
| Layer2 | Grad-CAM | 18.876 | 15.207 | 3.669 | 4.673 | 45.171 | Layer23 | Grad-CAM | 11.226 | 14.503 | -3.277 | -3.696 | 37.526 |
| | Grad-CAM++ | 19.779 | 14.901 | 4.879 | 7.165 | 44.16 | | GradCAM++ | 20.927 | 10.394 | 10.533 | 12.468 | 45.072 |
| | XGrad-CAM | 19.804 | 13.162 | 6.642 | 7.649 | 46.257 | | XGrad-CAM | 18.142 | 9.216 | 8.926 | 12.036 | 53.162 |
| | Eigen-CAM | 47.725 | 8.716 | 39.009 | 26.380 | 54.459 | | Eigen-CAM | 42.230 | 7.276 | 34.954 | 25.417 | 40.761 |
| | Layer-CAM | 45.826 | 7.660 | 38.166 | 26.864 | 52.579 | | Layer-CAM | 39.160 | 5.882 | 33.278 | 25.656 | 51.396 |
| | CAMERAS | 47.048 | 7.314 | 38.332 | 27.236 | 51.852 | | CAMERAS | 39.994 | 6.972 | 33.022 | 26.008 | 51.398 |
| | Relevance-CAM | 48.854 | 8.949 | 39.909 | 26.734 | 47.163 | | Relevance-CAM | 32.180 | 8.487 | 23.693 | 19.995 | 46.068 |
| | **Holistic-CAM** | **53.825** | **10.912** | **42.913** | **27.650** | **55.745** | | **Holistic-CAM** | **45.001** | **6.969** | **38.032** | **26.252** | **58.155** |

**Table 2: Ablation Study on Primary Modules.** Layer4 and Layer2 represent the last convolutional layer and the intermediate layer in ResNet, respectively; "P" represents positive gradient enhancement and "R" represents fundamental scale denoising. ✗ and ✓ denote module is removed or retrained separately. The best results are shown in **bold** and the second one is underlined.

| | P | F | Ins ↑ | Del ↑ | Over-all ↑ | EPG ↑ | | P | F | Ins ↑ | Del ↑ | Over-all ↑ | EPG ↑ |
|---|---|---|---|---|---|---|---|---|---|---|---|---|---|
| Layer4 | ✗ | ✗ | 54.418 | 8.878 | 45.540 | 52.845 | Layer2 | ✗ | ✗ | 47.124 | 7.926 | 39.828 | 51.734 |
| | ✓ | ✗ | 53.339 | 8.807 | 45.532 | 53.765 | | ✓ | ✗ | 50.066 | 8.414 | 41.652 | 55.213 |
| | ✗ | ✓ | 55.164 | 9.101 | **46.064** | 56.803 | | ✗ | ✓ | 53.001 | 11.021 | 41.980 | 52.922 |
| | ✓ | ✓ | 55.099 | 9.067 | 46.032 | **57.102** | | ✓ | ✓ | 53.878 | 10.904 | **42.925** | **55.577** |

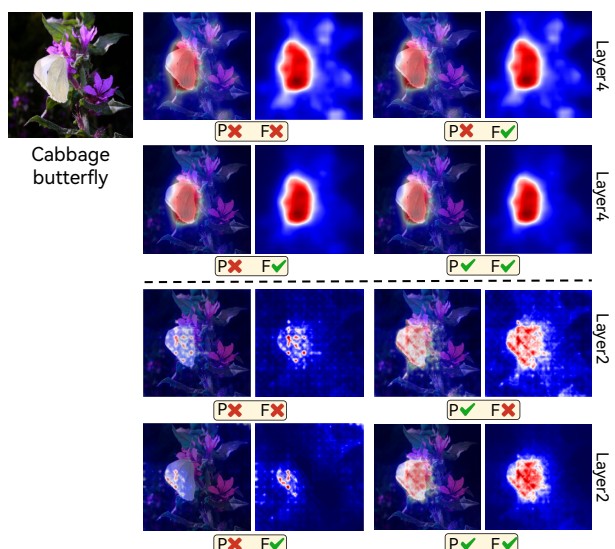

**Fig 8. Visualization of Ablation Study.** "P" represents positive gradient enhancement and "F" represents fundamental scale denoising.

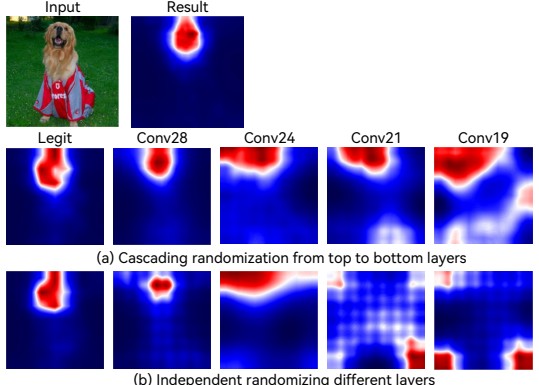

**Fig 9.** Sanity Checking of Cascading randomization and Independent randomization

## 5 RELATED WORK

Generally, attribution methods can be classified into two main categories: perturbation-based methods and backpropagation-based methods.

**Perturbation-based Attribution Methods:** These methods are intended to generate attribution maps by making pixel-level deliberate modifications to the input images and then associating them with the varying output results. Examples of perturbation-based methods include EMP [28], I-GOS [29], RISE [24], and Score-CAM [12]. The EMP [28] recommends using modified images to identify the precise area of interest for the predictor, while I-GOS [29] enhances EMP to achieve more effective convergence. RISE [24] produces attribution maps through the allocation of weights to random perturbation masks that align with the alterations in the output score. Score-CAM [12] employs the feature maps in a certain layer as the initial mask andassesses the saliency by examining the fluctuations in predicted output before and after masking. Nonetheless, these approaches often require extensive experimentation to

identify effective perturbation combinations, leading to significant computational inefficiency.

**Backpropagation-based Attribution Methods:** These techniques have shown enhanced efficacy by necessitating only a limited number of forward and backward propagation iterations. Generally, they can be classified as gradient-based methods and activation-based methods. Gradient-based methods [30] utilize pixel-level attribution to offer detailed and high-resolution interpretations for each layer. However, due to gradient noise, they often produce attribution maps of lower quality [18]. Activation mapping methods [8, 9], on the other hand, use back-flowed gradient information in conjunction with activation maps to generate faithful yet highly narrow interpretations at the final layer of CNNs. Other approaches such as Layer-CAM [14], NormGrad [15], and Relevance-CAM [18] aim to provide fine-grained interpretations within the broader feature spaces of shallow layers. Nevertheless, they face challenges in producing clear attribution maps near the output layer. CAMERAS [16] and MSG-CAM [17] employ multi-scale fusion techniques to enhance the resolution of interpretations within the output layer. However, their effectiveness in improving resolution in shallow layers remains limited, and they also encounter issues related to unexpected noise resulting from the integration of higher-resolution information.

## 6 CONCLUSION

In this paper, we introduce a novel attribution method called Holistic-CAM for fine-grained and sanity-preserving visual interpretation in the holistic stage of CNNs. The proposed attribution method is robust to the issues that other attribution methods face, such as the limited resolution in deep layers as well as unclear and faithless attributions in shallow layers. Our Holistic-CAM is capable of elucidating the model-central attention due to its full-stage high-resolution interpretation. Massive experiments demonstrate that Holistic-CAM outperforms the prevalent visual interpretation methods on common-used benchmarks. In the future, we will delve into the interpretation of multi-modal deep learning models.

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
