# OpenReview forum: "Holistic-CAM: Ultra-lucid and Sanity Preserving Visual Interpretation in Holistic Stage of CNNs"
_acmmm.org/ACMMM/2024/Conference — MM2024 Poster_

### Official Review · Reviewer_yYth · 2024-05-26

**Rating:** 4
**Confidence:** 2

**Summary:**

The article introduces a method called Holistic-CAM for high-definition interpretation in the holistic stage of CNNs. The approach initially employs weighted positive gradients to ensure the sanity of interpretations in the shallow layers and utilizes multi-scale fusion to enhance the resolution across the holistic stage. Fundamental scale denoising is then applied to eliminate the faithless attributions that arise from the fusion of large-scale components. This method is capable of rendering fine-grained and faithful attributions for CNNs from the shallow to the deep layers.

**Strengths:**

The authors have introduced an innovative method, which leverages weighted positive gradients to ensure the rationality of the attribution method for interpretations in the shallow layers of CNNs. They have creatively employed multi-scale fusion to enhance the resolution of the attribution method across the holistic stage. Additionally, to address the issue of false attributions introduced by multi-scale fusion, they have adopted fundamental scale denoising to eliminate these unfaithful attributions. Based on the proposed attribution method, the authors conducted ablation studies that demonstrated the contributions of two main modules: fundamental scale denoising and positive gradient enhancement, to the overall performance. The authors have validated their model's superiority over existing methods through tests such as deletion and insertion, and energy-based point games.

**Limitations:**

The article does not explicitly address whether the processes of multi-scale fusion and denoising will significantly increase the computational burden. Although Holistic-CAM is sensitive to model parameters, the paper does not provide a detailed discussion of its performance variance across different CNN architectures. Additionally, the applicability and effectiveness of Holistic-CAM across various types of CNN tasks are not discussed.

**Suitability:**

2

---

### Official Review · Reviewer_LmQ2 · 2024-05-29

**Rating:** 5
**Confidence:** 2

**Summary:**

The paper introduces a novel attribution method for high-definition visual interpretation throughout the entire depth of convolutional neural networks (CNNs). The authors address the shortcomings of existing methods that only focus on the final convolutional layer, resulting in narrow and concentrated interpretations. Their approach employs weighted positive gradients to maintain interpretability in shallow layers, and they propose a fundamental scale denoising technique to eliminate unfaithful attributions when using multi-scale fusion to enhance resolution. Extensive experiments demonstrate that Holistic-CAM outperforms state-of-the-art methods on common-used benchmarks.

**Strengths:**

The paper tackles a significant problem in CNN interpretation, proposing a novel method to enhance visual interpretations throughout all stages of the network. The introduction effectively motivates the research by clearly highlighting the limitations of existing methods. The proposed method is thoroughly explained and supported with detailed descriptions and illustrative figures. Additionally, the experimental results convincingly demonstrate the superiority of Holistic-CAM over state-of-the-art methods across various benchmarks, validating the effectiveness of the approach.

**Limitations:**

Previous methods have employed positive gradients to improve interpretations, but the paper demonstrates that their practical impacts remain inadequate in output layers. The authors should explore and clarify the detailed differences to understand why these previous approaches fall short.

**Suitability:**

2

---

### Official Review · Reviewer_pohF · 2024-06-09

**Rating:** 4
**Confidence:** 2

**Summary:**

The paper introduces Holistic-CAM, a visual interpretation method for CNNs that addresses limitations of existing methods focusing solely on the final convolutional layer. Holistic-CAM uses weighted positive gradients and multi-scale fusion to provide high-resolution and clear visual interpretations across all CNN layers, ensuring faithful attributions and reducing noise from larger-scale components. Extensive experiments on ImageNet-1K demonstrate that Holistic-CAM outperforms current state-of-the-art methods on benchmarks such as deletion and insertion, energy-based point game, and remove and debias.

**Strengths:**

The paper is well-structured and easy to follow. Holistic-CAM provides high-resolution and clear visual interpretations across all CNN layers by employing weighted positive gradients and multi-scale fusion, ensuring accurate attributions and minimizing noise.

**Limitations:**

1.	Some details are unclear. For example, in Line 455, the method for obtaining the fundamental scale component is not well-explained. leaving readers unsure about the specific steps involved in this process. Clarifying this and other procedural aspects would greatly enhance the paper’s comprehensibility.

2.	In Figure 1 of the supplementary material, it appears that even in the first layer, the proposed Holistic-CAM is able to locate the target object, which seems unreasonable. Typically, the shallow layers of deep networks should focus on edges and basic features, not on locating the entire object. This discrepancy raises questions about the accuracy and interpretability of the visualizations provided by Holistic-CAM. Further clarification on how Holistic-CAM achieves such detailed interpretations in the early layers would be beneficial.

3.	In Table 1, why does the proposed Holistic-CAM perform significantly worse than other methods (i.e., Eigen-CAM, Layer-CAM, and CAMERAS) for the results of Layer 2 in ResNet-50? It would be better to provide more explanations.

4.	In Figure 8, what is the difference between the second column of row 1 and the first column of row 2? They appear to be the same figure.

5.	The experiments primarily focus on heavyweight models like ResNet-50 and VGG-16. Is it possible to apply the proposed method to lightweight models such as MobileNetV2?

6.	The proposed method mainly focuses on CNNs. Is it possible to extend this approach to vision transformers?

Reference:
[1] Mobilenetv2: Inverted residuals and linear bottlenecks. CVPR 2018.

**Suitability:**

3

---

### Official Review · Reviewer_Q5he · 2024-06-10

**Rating:** 4
**Confidence:** 2

**Summary:**

This paper introduces Holistic-CAM, a new method for visual interpretation of convolutional neural networks (CNN). Holistic-CAM has made some progress in the field of visual interpretation of CNNs, solving the main limitations of existing methods. Its ability to provide high-resolution and reliable interpretation across the entire network is a significant advantage, however, future work is needed to address the increased complexity and potential challenges in generalizing and processing large-scale components.

**Strengths:**

Traditional methods such as Grad-CAM and its variants focus on the final convolutional layer, resulting in a narrow interpretation range and low resolution. These methods also produce confusing and disloyal attributions at a shallow level. Holistic-CAM aims to provide high-resolution and sanity-preserving visual interpretation of the entire CNN. Weighted positive gradients are utilized to process shallow layers and multi-scale fusion is used to provide high-resolution interpretation of all layers. Holistic-CAM solves the shortcomings of only focusing on the final convolutional layer by considering the entire CNN, providing a more detailed and accurate explanation.

**Limitations:**

Multi-scale fusion and weighted positive gradient techniques increase the complexity of implementation and may require more computational resources than simpler methods.
While this method shows promise on ImageNet-1k, it would be more beneficial to observe its performance on a wider range of datasets and tasks to ensure its general applicability.
The article discusses the limitations of using channel weights and suggests using element weights as a solution. However, numerical analysis of gradient variance may be too simplistic to cover all edge cases.
The comparative methods are all methods from a few years ago, which are too old. They do not show the relevant methods in the past two years. Whether this research field is still meaningful.

**Suitability:**

2

---

### Official Review · Reviewer_FGxH · 2024-06-26

**Rating:** 5
**Confidence:** 2

**Summary:**

The paper proposes a novel visual interpretation method called Holistic-CAM for convolutional neural networks (CNNs). The key contributions are:

Weighted positive gradients are used to ensure the sanity and robustness of interpretations in shallow layers of the CNN.
A multi-scale fusion scheme is leveraged to improve the resolution of the attribution maps across the holistic stage of the CNN.
A fundamental scale denoising technique is introduced to eliminate unfaithful attributions caused by fusing larger-scale components.

**Strengths:**

The proposed Holistic-CAM method is able to simultaneously provide fine-grained and faithful attribution maps from the shallow to deep layers of the CNN.

Strengths of the paper:

The method addresses the key limitations of existing CAM-based interpretation methods, which tend to generate low-resolution and unfaithful attributions, especially in the shallow layers.
The technical innovations, including weighted positive gradients, multi-scale fusion, and fundamental scale denoising, effectively improve the quality and fidelity of the visual interpretations.
Extensive experiments on benchmark datasets demonstrate the superior performance of Holistic-CAM compared to state-of-the-art methods.

**Limitations:**

Limitations and weaknesses:

The paper does not provide a comprehensive theoretical analysis or justification for the design choices made in the Holistic-CAM method.
The computational complexity of the multi-scale fusion and fundamental scale denoising components may limit the practical applicability of the method, especially for real-time or resource-constrained scenarios.
The paper lacks a thorough discussion of the potential limitations or failure cases of the Holistic-CAM method, as well as avenues for future improvements.
The paper does not explore the sensitivity of Holistic-CAM to different CNN architectures or domains beyond image classification tasks.

**Suitability:**

2

---

### Meta-Review · Program_Chairs · 2024-07-13

**Recommendation:** Accept (Poster)
**Confidence:** 5

**Metareview:**

The paper introduces Holistic-CAM, a visual interpretation method for convolutional neural networks. The method makes three key contributions: Firstly, it uses weighted positive gradients to ensure the accuracy and reliability of interpretations in shallow layers of the CNN. Secondly, it employs a multi-scale fusion scheme to enhance the resolution of attribution maps throughout the holistic stage of the CNN. Lastly, it introduces a fundamental scale denoising technique to eliminate inaccurate attributions caused by fusing larger-scale components.  The paper is well-written.  The experiment demonstrates the efficacy of the proposed approach. This paper can provide insights into the visual interpretation of deep models.

All four reviewers have given a positive rating to this paper. After carefully reviewing the paper, the authors' rebuttal, and both sets of reviewer comments (before and after the rebuttal), I recommend accepting this paper.

The reviewer, **FGxH**, has raised several additional questions in the Final Rating Justification:
> What is the theoretical foundation or intuition behind the choice of weighted positive gradients and the specific design of the multi-scale fusion and fundamental scale denoising components? How can the computational efficiency of Holistic-CAM be improved to make it more practical for real-world applications? What are the potential limitations or failure cases of Holistic-CAM, and how can the method be further improved to address these issues? How does Holistic-CAM perform on a wider range of CNN architectures and task domains, beyond the image classification tasks explored in the paper?

Solving these questions will enhance the quality of this paper. I hope the authors can include explanations and discussions in the revised version of the paper.